# NLRC4, ASC and Caspase-1 Are Inflammasome Components That Are Mediated by P2Y_2_R Activation in Breast Cancer Cells

**DOI:** 10.3390/ijms21093337

**Published:** 2020-05-08

**Authors:** Hana Jin, Hye Jung Kim

**Affiliations:** Department of Pharmacology, College of Medicine, Institute of Health Sciences, Gyeongsang National University, Jinju 52727, Korea; hanajin.kr@daum.net

**Keywords:** P2Y2 receptor, NLRC4 inflammasome, radiotherapy-resistant, breast cancer, tumor progression

## Abstract

The inflammasomes are reported to be associated with tumor progression. In our previous study, we determined that extracellular ATP enhances invasion and tumor growth by inducing inflammasome activation in a P2Y purinergic receptor 2 (P2Y_2_R)-dependent manner. However, it is not clear which inflammasome among the diverse complexes is associated with P2Y_2_R activation in breast cancer. Thus, in this study, we determined which inflammasome components are regulated by P2Y_2_R activation and are involved in tumor progression in breast cancer cells and radiotherapy-resistant (RT-R)-breast cancer cells. First, we found that NOD-, LRR-, and pyrin domain-containing protein 3 (NLRP3); NLR family caspase activation and recruitment domain (CARD) containing 4 (NLRC4); apoptosis-associated speck-like protein containing a CARD complex (ASC); and caspase-1 mRNA levels were upregulated in RT-R-MDA-MB-231 cells compared to MDA-MB-231 cells, whereas tumor necrosis factor-α (TNF-α) or ATP treatment induced NLRC4, ASC, and caspase-1 but not NLRP3 protein levels. Moreover, TNF-α or ATP increased protein levels of NLRC4, ASC, and caspase-1 in a P2Y_2_R-dependent manner in MDA-MB-231 and RT-R-MDA-MB-231 cells. In addition, P2Y_2_R activation by ATP induced the secretion of IL-1β and VEGF-A, as well as invasion, in MDA-MB-231 and RT-R-MDA-MB-231 cells, which was inhibited by NLRC4, ASC, and caspase-1 small interfering RNA (siRNA). Taken together, this report suggests that P2Y_2_R activation by ATP induces tumor invasion and angiogenesis through inflammasome activation, specifically by regulating the inflammasome components NLRC4, ASC, and caspase-1.

## 1. Introduction

Inflammation indicates that the body is fighting something harmful and trying to heal itself. In response to tissue damage, leukocytes such as monocytes, neutrophils, and eosinophils are recruited and infiltrate the site of injury, where the leukocytes begin to amplify a multifactorial network of chemical signals to heal the damaged tissue. In addition to cell proliferation and tissue recovery, inflammation is also responsible for eliminating dead cells and debris. These physiological inflammatory responses are self-restricted and are terminated after all harmful agents are discarded or repair is completed [1,2]. However, Virchov identified a correlation between inflammation and cancer for the first time and recognized the process as one of the predisposing conditions for tumor development [3]. Uncontrolled chronic inflammation can induce malignant cell growth and initiation of tumors in the surrounding tissue through the constant production of growth factors and reactive oxygen and nitrogen species that result in permanent genomic alterations by interacting with DNA in the proliferating epithelium [4,5]. Furthermore, inflammation plays an important role in tumor promotion, malignant conversion, and metastasis because various inflammatory regulators, including cytokines, chemokines, growth factors, free radicals, and prostaglandins, as well as proteolytic enzymes stimulate tumor development [6]. These inflammatory mediators are produced and secreted by diverse types of cells in the tumor microenvironment, including immune cells (leukocytes, lymphocytes, dendritic cells, and natural killer cells), fibroblasts, adipocytes, endothelial cells, and tumor cells [7]. Some factors directly affect tumor cells and stimulate cell proliferation, inhibit cell death, and promote oncogenic mutations, whereas other factors show protumoral activities by acting on other components of the tumor microenvironment [8,9].

Interleukin (IL)-1, which is abundant in the tumor microenvironment, is one of the crucial proinflammatory cytokines that can control the growth and invasiveness of tumor cells [10]. IL-1 is classified into two agonistic forms: IL-1α is active as an intracellular protein, and IL-1β is active as a secreted protein. Increased IL-1β levels in tumors and serum are associated with increased tumor grade and enhanced invasion in several cancers, including breast cancer, and are correlated with poor prognosis [11,12,13,14,15,16]. Eventually, the major cause of mortality from cancer is disease recurrence and metastasis, which is the spreading of primary tumor cells to distant sites to form secondary tumors. IL-1β has been implicated in tumor growth, angiogenesis, invasiveness, relapse, and progression [17,18,19]. The production and secretion of the mature/active form of IL-1β, a potent protumoral regulator, is mediated by inflammasome activation [20]. The inflammasome is a multimolecular complex composed of a sensor receptor, the adaptor apoptosis-associated speck-like protein containing a caspase recruitment domain (apoptosis-associated speck-like protein containing a CARD complex (ASC)), and pro-caspase-1. The inflammasome sensors are classified into nucleotide-binding domain-like receptors (NLRs), absent in melanoma 2-like (AIM2) receptors, and the recently identified pyrin. Once activated by environmental stimuli, pathogen-associated molecular patterns (PAMPs), and damage-associated molecular patterns (DAMPs), these receptors assemble the inflammasome complex and activate the cysteine protease caspase-1. Active caspase-1 is responsible for the cleavage of pro-IL-1β and pro-IL-18 proteins into their mature active forms, ultimately resulting in their secretion [20]. Especially in breast cancer, nucleotide-binding oligomerization domain (NOD)-, leucine-rich repeat (LRR)-, and pyrin domain-containing protein 1 (NLRP1); NLRP3; NLR family CARD domain containing 4 (NLRC4); and AIM2 inflammasome complex proteins show pro- or antitumoral activities with increased expression. NLRP1 expression is higher in primary breast cancer tissue than in adjacent noncancerous tissue and is associated with lymph node metastasis, tumor-node-metastasis (TNM) stage, and Ki-67 levels. Moreover, NLRP1 promotes breast cancer cell proliferation, migration, and invasion by inducing the epithelial–mesenchymal transition (EMT) [21]. The NLRC4 inflammasome mediates the expression of adipocyte-mediated vascular endothelial growth factor A and angiogenesis, which accelerates the progression of breast cancer [22]. The NLRP3 inflammasome is the most extensively studied inflammasome, and activation contributes to immune system dysfunction and breast cancer metastasis by mediating the secretion of IL-1β [23]. AIM2 expression suppresses the proliferation and tumorigenicity of human breast cancer cells as well as mammary tumor growth in an orthotopic tumor model [24].

Extracellular adenosine triphosphate (ATP) secretion is triggered by diverse cellular stimuli, and it is a key DAMP that induces NLRP3 inflammasome formation by signaling through the purinergic P2X_7_ cell surface receptor (P2X_7_R) [25,26,27]. Interestingly, ATP is highly accumulated in the tumor microenvironment by release from both damaged cells and tumor cells and is involved in tumor progression [28,29]. In addition, ATP activates assembly of the inflammasome complex in cells in the tumor microenvironment [30]. Our previous results suggested that highly metastatic breast cancer cells MDA-MB-231 cells released high levels of ATP, and the subsequent activation of P2Y purinergic receptor 2 (P2Y_2_R) promoted invasion and tumor growth by inducing inflammasome activation [31]. Interestingly, radiotherapy-resistant (RT-R)-MDA-MB-231 cells derived from MDA-MB-231 cells by repeated irradiation released higher levels of ATP than MDA-MB-231 cells and showed a more increased inflammasome activation in a P2Y_2_R-dependent manner [31]. Radiotherapy is a crucial treatment in cancer therapy, however, tumor relapse following radiotherapy is common problem and breast cancer-acquired resistance for radiotherapy leads to therapeutic failure due to their aggressive properties. Therefore, the study to clarify the molecular mechanism involved in tumor progression and aggressiveness of RT-R-breast cancer cells is needed. As mentioned above, inflammasomes play a critical role in tumor progression, but it is not well known yet which inflammasome is associated with P2Y_2_R activation in breast cancer. Thus, in this study, we aimed to determine which inflammasome components are regulated by P2Y_2_R activation and are involved in tumor progression in breast cancer cells and RT-R breast cancer cells.

## 2. Results

### 2.1. Compared to MDA-MB-231 Cells, RT-R-MDA-MB-231 Cells Exhibited Increased mRNA Levels of NLRP3, NLRC4, ASC, and Cleaved Caspase-1, and NLRC4, ASC, and Cleaved Caspase-1, but Not NLRC3, Were Induced by TNF-α and ATP in Both Breast Cancer Cells

First, we compared the expression levels of inflammasome components that were reported to be expressed in MDA-MB-231 and RT-R-MDA-MB-231 breast cancer cells. Interestingly, NLRP3, NLRC4, ASC, and cleaved caspase-1 mRNA levels were increased in RT-R-MDA-MB-231 cells compared to MDA-MB-231 cells. However, NLRP1 and AIM2 showed lower expression levels in RT-R-MDA-MB-231 cells than in MDA-MB-231 cells. (Figure 1A). In our previous study, RT-R-MDA-MB-231 cells released higher levels of IL-1β, which is secreted by inflammasome activation, than MDA-MB-231 cells, and IL-1β was related to tumor progression in MDA-MB-231 and RT-R-MDA-MB-231 cells, suggesting its involvement in tumor aggressiveness. Thus, we chose the inflammasome components NLRP3, NLRC4, ASC, and caspase-1, which displayed increased mRNA expression levels, and then determined their protein levels in MDA-MB-231 and RT-R-MDA-MB-231 cells in response to treatment with tumor necrosis factor-α (TNF-α) or ATP, which accumulate in the tumor microenvironment. As shown in Figure 1B–E, the protein levels of NLRC4, ASC, and cleaved caspase-1, but not NLRP3, were significantly increased by treatment with TNF-α or ATP in both MDA-MB-231 and RT-R-MDA-MB-231 cells.

### 2.2. Expression of the Inflammasome Components NLRC4, ASC, and Cleaved Caspase-1 Were Induced by TNF-α or ATP in a P2Y_2_R-Dependent Manner in MDA-MB-231 and RT-R-MDA-MB-231 Cells

Next, we examined whether ATP-mediated P2Y_2_R activation is related to the induction of inflammasome components in MDA-MB-231 and RT-R-MDA-MB-231 cells using apyrase, which is an enzyme that rapidly hydrolyzes extracellular nucleotides, or P2Y_2_R small interfering RNA (siRNA). First, we found that the TNF-α-induced increased protein expression levels of NLRC4, ASC, and cleaved caspase-1 were markedly decreased in the presence of apyrase in both MDA-MB-231 and RT-R-MDA-MB-231 cells (Figure 2A,B). Moreover, P2Y_2_R knockdown using P2Y_2_R siRNA (Figure 2C) also notably suppressed the TNF-α or ATP treatment-induced increase in inflammasome component protein expression in both MDA-MB-231 and RT-R-MDA-MB-231 cells (Figure 2D–G). However, NLRP3 expression was not enhanced by treatment with TNF-α or ATP and was not affected by apyrase treatment or P2Y_2_R knockdown (Figure 2). These results suggest that P2Y_2_R activation by ATP released from breast cancer cells regulates the expression levels of NLRC4, ASC, and cleaved caspase-1 but not NLRP3 in both MDA-MB-231 and RT-R-MDA-MB-231 cells.

### 2.3. ATP-Induced Expression of the Inflammasome Components NLRC4, ASC, and Cleaved Caspase-1 Enhanced the Secretion of IL-1β and VEGF-A, as well as Invasion, in MDA-MB-231 and RT-R-MDA-MB-231 Cells in a P2Y_2_R-Dependent Manner

In our previous study, we determined that ATP treatment-induced activation of the inflammasome increased the secretion of IL-1β in a P2Y_2_R-dependent manner, ultimately resulting in increased tumor invasion and progression [31]. Thus, we investigated whether the ATP-enhanced inflammasome components are involved in IL-1β secretion and consequently related to tumor progression, including invasion and angiogenesis, in both breast cancer cells, and whether the effects are P2Y_2_R-dependent. First, we found that the IL-1β and vascular endothelial growth factor (VEGF)-A secretion levels were higher in RT-R-MDA-MB-231 cells than in MDA-MB-231 cells and were markedly induced in response to ATP treatment, and this effect was significantly abolished by P2Y_2_R knockdown in both MDA-MB-231 and RT-R-MDA-MB-231 cells. Interestingly, P2Y_2_R siRNA transfection decreased the levels of IL-1β and VEGF-A produced by RT-R-MDA-MB-231 cells in the absence of ATP (Figure 3A,B). Furthermore, RT-R-MDA-MB-231 cells showed high invasiveness compared to MDA-MB-231 cells, and the enhanced invasive ability of MDA-MB-231 and RT-R-MDA-MB-231 cells by ATP treatment was significantly suppressed by P2Y_2_R knockdown in both MDA-MB-231 and RT-R-MDA-MB-231 cells (Figure 3C,D).

Finally, we confirmed that P2Y_2_R activation by ATP mediated IL-1β and VEGF-A production and subsequent invasiveness of breast cancer cells through activation of NLRC4, ASC, and caspase-1 of inflammasome components. MDA-MB-231 and RT-R-MDA-MB-231 cells were transfected with siRNAs targeting each inflammasome components (Figure 4A), and then IL-1β secretion, VEGF-A production, and invasion were measured. Figure 4B,C shows that ATP treatment-induced increases in IL-1β secretion and VEGF-A production levels were notably suppressed by siRNA knockdown of NLRC4, ASC, and caspase-1, but not by NLRP3 siRNA knockdown in both breast cancer cells. Moreover, the enhanced invasive ability of MDA-MB-231 and RT-R-MDA-MB-231 cells by ATP treatment was also significantly decreased by NLRC4, ASC, and caspase-1 siRNA, but not NLRP3 siRNA (Figure 4D,E). These results suggest that P2Y_2_R activation by ATP induces tumor invasion and angiogenesis through inflammasome activation, particularly by regulating the inflammasome components NLRC4, ASC, and caspase-1.

## 3. Discussion

Under pathological conditions, the extracellular nucleotide ATP acts as a proinflammatory danger signal and activates the inflammasomes, specifically NLRP3, through binding to P2X_7_R [26]. In our previous study, we revealed that compared to MDA-MB-231 cells, RT-R-MDA-MB-231 cells display enhanced malignant properties involved in tumor progression, such as migration, invasion, adhesion to endothelial cells, and chemoresistance [32]. In addition, we found that RT-R breast cancer cells release increased levels of ATP and show enhanced invasion and tumor growth by inducing inflammasome activation in a P2Y_2_R-dependent manner [31]. Therefore, we hypothesized that the increased inflammasome components take part in the malignancy of RT-R breast cancer cells by mediating the release of mature IL-1β. Inflammasome complexes are expressed in various tumors or cancer cells and regulate tumor development by maturing and secreting IL-1β into the tumor microenvironment [20]. However, these complexes are known to show different expressions and functions according to tumor types. In addition, it is still unknown which inflammasome is regulated by P2Y_2_R activation; thus, we aimed to investigate which inflammasome complex is regulated by P2Y_2_R activation. Accordingly, in this study, we compared the mRNA expression levels of inflammasome components that are reported to be expressed in breast cancer and have protumoral activities [33] in MDA-MB-231 and RT-R-MDA-MB-231 cells. As a result, NLRP3, NLRC4, ASC, and cleaved caspase-1 showed higher mRNA expression levels in RT-R-MDA-MB-231 cells than in MDA-MB-231 cells (Figure 1A). Interestingly, NLRC4, ASC, and cleaved caspase-1 protein levels but not NLRP3 were induced in response to TNF-α or ATP treatment in both MDA-MB-231 and RT-R-MDA-MB-231 cells. Although NLRP3 mRNA expression levels were increased in RT-R-MDA-MB-231 cells compared to MDA-MB-231 cells, the protein levels were not very different and were not responsive to TNF-α or ATP treatment in either type breast cancer cell (Figure 1B–E). Moreover, NLRC4, ASC, and caspase-1 protein levels were induced in response to TNF-α or ATP treatment in a P2Y_2_R-dependent manner (Figure 2), suggesting that the ATP released by TNF-α treatment activates P2Y_2_R and regulates inflammasome expression in breast cancer cells. In addition, we found that RT-R-MDA-MB-231 cells showed an increased level of IL-1β and VEGF-A production compared to that of MDA-MB-231 cells, which was enhanced by stimulation with ATP in a P2Y_2_R-dependent manner (Figure 3A,B). The mature secreted IL-1β is tightly involved in human tumor progression, tumor angiogenesis, invasion, and metastasis [34]. As well as IL-1β and VEGF-A secretion, we observed that the invasion ability was also highly enhanced in RT-R-MDA-MB-231 cells compared with MDA-MB-231 cells in response to ATP treatment (Figure 3C). Increased inflammasome activation, which was observed by IL-1β secretion, and as a result, the induced VEGF-A production and invasion ability in RT-R-MDA-MB-231 were significantly reduced by NLRC4, ASC, and caspase-1 siRNA (Figure 4). These results suggest that NLRC4, ASC, and caspase-1 are inflammasome components that are mediated by P2Y_2_R activation in breast cancer cells. Even though this is a common finding observed in both MDA-MB-231 and RT-R-MDA-MB-231, it is interesting that these inflammasome components NLRC4, ASC, and caspase-1 were found to be more induced in RT-R-MDA-MB-231 than in MDA-MB-231 and were involved in the induced IL-1β secretion, VEGF-A production, and invasion in RT-R-MDA-MB-231 cells.

Notably, NLRP1 and AIM2 mRNA expression levels were lower in RT-R-MDA-MB-231 cells than in MDA-MB-231 cells (Figure 1A). Mounting evidence has reported that inflammasome signaling is tightly related to human tumor progression; however, the function of the inflammasome in tumor development remains controversial. According to reports, increased AIM2 inflammasome expression induced by interferon-γ (IFNγ) promotes apoptosis via the mitochondrial pathway and the regulation of proapoptotic proteins in MCF-7 breast cancer cells [35]. In addition, the NLRP3 inflammasome enhances antitumor function via stimulation of dendritic cells [36]. In contrast, the IL-1β/IL-1R1/β-catenin signaling pathway that regulates c-MYC, cyclin D1 (CCDN1), SNAIL1, and matrix metallopeptidases (MMP) 2 expression promotes proliferation, migration, and invasion in breast cancer [37]. Moreover, IL-1β promotes breast ductal cancer cell invasiveness by activating extracellular signal regulated kinase (ERK) 1/2 [38]. Similarly, our previous and present work revealed that inflammasome activation enhanced proliferation, migration, and invasion in highly metastatic breast cancer cells. In particular, P2Y_2_R activation by ATP released from breast cancer cells induced tumor progression by regulating the inflammasome components NLRC4, ASC, and caspase-1. Even though the role of the inflammasome components NLRP1 and AIM2 in tumor progression was not clarified in this study because of the low expression levels in RT-R-MDA-MB-231 cells compared to MDA-MB-231 cells, it is necessary to further study the function and role of these two components in tumor progression.

It has been reported that IL-1β induces VEGF expression and production in several cell types [39,40]. VEGF refers to the originally identified dimeric glycoprotein termed VEGF-A and the VEGF-related polypeptide family, including VEGF-B, VEGF-C, VEGF-D, and placental growth factor. VEGF-A was first shown to be an endothelial growth factor and a regulator of vascular permeability. In the tumor microenvironment, VEGF-A is produced by hypoxic tumor cells, endothelial cells, infiltrating myeloid cells, and tumor-associated macrophages [41,42]. To initiate signaling cascades, VEGF family proteins bind to the receptor tyrosine kinases vascular endothelial growth factor receptor 1 (VEGFR1), VEGFR2, and VEGFR3, and the in vivo angiogenic response to VEGF-A is mainly mediated through VEGFR2 activation [43]. VEGF-A-induced pathways closely regulate angiogenesis by coordinating endothelial cell proliferation, migration, and invasion, leading to the formation of lumen-containing structures; endothelial cell proliferation and survival are stimulated via the ERK [44] and phosphatidylinositol 3-kinase (PI3K)/AKT pathways [45]. Interestingly, siRNA-mediated P2Y_2_R knockdown alone suppressed IL-1β and VEGF-A production in MDA-MB-231 and RT-R-MDA-MB-231 cells in our study. In fact, both MDA-MB-231 and RT-R-MDA-MB-231 cells released ATP into the extracellular environment, even though RT-R-MDA-MB-231 cells released more ATP than MDA-MB-231 cells. Thus, ATP released from both breast cancer cells during the incubation time (24 h) to detect IL-1β and VEGF production could activate P2Y_2_R to induce IL-1β and VEGF-A production without the addition of exogenous ATP. This could be a possible reason why P2Y_2_R siRNA itself reduced IL-1β and VEGF-A production below the control level of both cells and even more so in RT-R-breast cancer cells.

## 4. Materials and Methods

### 4.1. Materials

Recombinant human TNF-α protein and human IL-1β/IL-1F2 Quantikine ELISA kits were purchased from R&D Systems (Minneapolis, MN, USA). ATP, apyrase, anti-β-actin (#A2066) antibody, and 4′,6-diamidine-2′-phenylindole dihydrochloride (DAPI) were obtained from Sigma-Aldrich (St. Louis, MO, USA). Anti-NLRP3 (#ab214185), anti-NLRC4 (#ab99860), and anti-cleaved caspase-1 (#ab179515) antibodies and human VEGF-A ELISA kits were purchased from Abcam (Cambridge, United Kingdom), and anti-ASC (#13833) antibody was obtained from Cell Signaling Technology (Beverly, MA, USA). TOPscript One-step RT PCR Drymix was purchased from Enzynomics (Daejeon, Korea). Negative control siRNA (control siRNA) and NLRP3, NLRC4, ASC, caspase-1, and P2Y_2_R siRNA were obtained from Bioneer (Daejeon, Korea), and Lipofectamine 3000 reagent, TRIzol reagent, and anti-P2Y_2_R (#PA1-46150) antibody were purchased from Thermo Fisher Scientific (Waltham, MA, USA). Hybond-P^+^ polyvinylidene difluoride membrane was obtained from GE Healthcare (Chicago, IL, USA), and Clarity Western enhanced chemiluminescence (ECL) substrate was purchased from Bio-Rad (Hercules, CA, USA). BD Matrigel basement membrane matrix (Matrigel) was obtained from BD Bioscience (Franklin Lakes, NJ, USA), and cell culture inserts for 24-well plates (8.0 µm, Translucent PET membrane) were purchased from Corning (Corning, NY, USA).

### 4.2. Establishment of RT-R-MDA-MB-231 Cells and Cell Culture

The human breast cancer cell line MDA-MB-231 was purchased from the Korea Cell Line Bank (Seoul, Korea). RT-R-MDA-MB-231 cells were established as described by Ko et al. [32]. Briefly, MDA-MB-231 cells were repeatedly irradiated with 2 Gy using a 6-MV photon beam that was produced by a linear accelerator (Clinac 21EX, Varian Medical Systems, Inc., Palo Alto, CA, USA) until a final dose (50 Gy) was achieved, which is a commonly used clinical regimen for radiotherapy in patients with breast cancer. RT-R-MDA-MB-231 cells were cultured in Roswell Park Memorial Institute (RPMI)-1640 supplemented with 10% fetal bovine serum (FBS), 100 IU/mL penicillin, and 10 µg/mL streptomycin, and incubated at 37 °C in a humidified atmosphere containing 5% CO_2_ and 95% air. MDA-MB-231 and RT-R-MDA-MB-231 cells were used within five passages.

### 4.3. Reverse Transcription PCR (RT-PCR)

RT-PCR was performed using TOPscript One-step RT PCR Drymix according to the manufacturer’s instructions. The primer sets used are described in Table 1. Thirty cycles of amplification were performed under the following conditions: melting at 95 °C for 30 s, annealing at the indicated temperature for 30 s, and extension at 72 °C for 45 s.

### 4.4. Protein Extraction and Western Blotting

For the isolation of protein extract from whole cells, cells were washed with ice-cold phosphate buffered saline (PBS) and lysed with radioimmunoprecipitation assay (RIPA) buffer (0.1% nonylphonoxypolyethoxypthanol-40 (NP-40) and 0.1% sodium dodecyl sulfate (SDS) in PBS) containing a protease inhibitor cocktail. The suspension was centrifuged at 13,000 rpm for 15 min, and then the supernatant (protein extract) was obtained. Approximately 40–80 μg aliquots of protein were subjected to 10% SDS-polyacrylamide gel electrophoresis and transferred onto membranes. The membranes were incubated with blocking reagent (5% nonfat milk in TBS-T) and then subsequently incubated with the indicated antibodies at 4 °C. The bound antibodies were detected with secondary antibodies and an ECL Western blotting detection reagent.

### 4.5. Gene Silencing with siRNA Transfection

Cells were transfected with 100 nM negative control siRNA (control siRNA) or NLRP3, NLRC4, ASC, caspase-1, or P2Y_2_R siRNA in serum-free medium using Lipofectamine 3000 for 4 h, and then the medium was replaced with fresh complete medium. The cells were starved with serum-free medium for 16 h and then treated with the indicated reagents. Gene silencing efficiency was determined by Western blot analysis.

### 4.6. Quantification of IL-1β and VEGF-A Secretion

To quantify the amounts of secreted IL-1β or VEGF-A, cell culture supernatants were assayed using ELISA kits according to the manufacturer’s protocol. Briefly, 200 μL of sample was added to a microplate and sequentially reacted with conjugate, substrate solution, and stop solution. The optical density of each well was measured at a wavelength of 450 nm using a microplate reader.

### 4.7. Matrigel Invasion Assay

For invasion assays, the upper chambers of inserts were coated with 100 μL of Matrigel (1 mg/mL), and endothelial cells (2 × 10^5^ cells) were added to the Matrigel-coated insert wells. Then, MDA-MB-231 or RT-R-MDA-MB-231 cells (2 × 10^5^ cells/500 μL) were added to each upper chamber in serum-free medium, and 500 μL of complete medium was added to the lower chamber. The invasion chambers were incubated for 24 h at 37 °C. The noninvaded cells remaining on the upper surface of the insert membranes were removed by scrubbing. The invaded cells that passed through the insert membrane were stained with DAPI and counted in three randomly selected field under a fluorescence microscope.

### 4.8. Statistical Evaluations

The treatment groups were compared using one-way analysis of variance (ANOVA) and Newman–Keuls post hoc tests. *p* < 0.05 was considered statistically significant. All data were evaluated for normality and homogeneity of variance and are expressed as the mean ± standard error of the mean (SEM).

## 5. Conclusions

In conclusion, in the present study, we demonstrated that P2Y_2_R, which is activated by ATP secreted from breast cancer cells, regulates inflammasome expression, especially the inflammasome components NLRC4, ASC, and caspase-1, ultimately resulting in increased tumor invasion and angiogenesis. As radiotherapy resistance occurs frequently in patients with breast cancer, and because RT-R breast cancer cells have more aggressive properties, regulation of the aggressiveness of RT-R breast cancer might be an important and specific approach in cancer therapy. Moreover, as mentioned in the Introduction, inflammation has two aspects—on one hand, it is a defense mechanism to protect body from the damage, and on the other hand, it also plays an important role in tumor promotion, malignant conversion, and metastasis. Thus, to clarify the specific inflammasome components that are involved in the tumor progression and recurrence and to regulate those components could be an essential trial for cancer treatment. In this respect, our results suggest for the first time that P2Y_2_R is involved in activation of the NLRC4 inflammasome, which is composed of NLRC4, ACS, and caspase-1, and highlight the importance of regulating P2Y_2_R activity to achieve a good prognosis in patients with RT-R tumors (Figure 5).

## Figures and Tables

**Figure 1 ijms-21-03337-f001:**
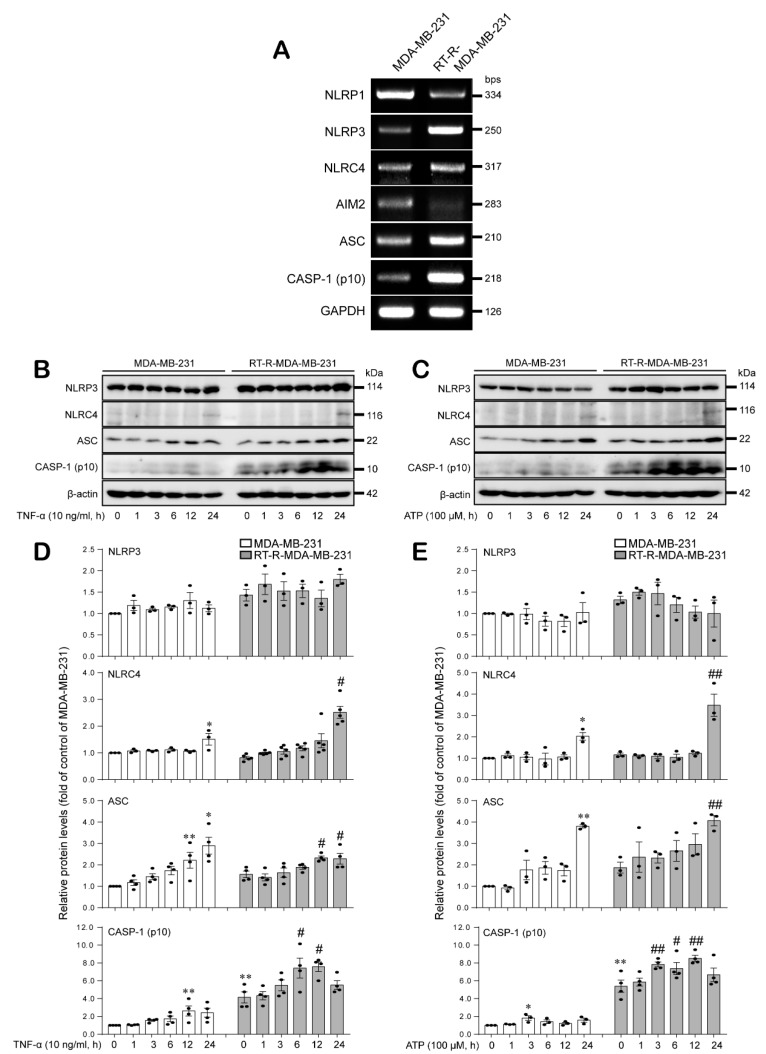
Expression level of inflammasome components and induction of these components in response to tumor necrosis factor-α (TNF-α) and ATP in both MDA-MB-231 and radiotherapy-resistant (RT-R)-MDA-MB-231 cells. (**A**) Total RNA was collected, and the mRNA levels of inflammasome components (nucleotide-binding domain-like receptor NOD-, LRR-, and pyrin domain-containing protein 1 (NLRP1); NLRP3; NLRP6; nucleotide-binding domain-like receptor (NLR) family CARD domain containing 4 (NLRC4); absent in melanoma 2-like (AIM2); apoptosis-associated speck-like protein containing a CARD complex (ASC); and cleaved caspase-1) and glyceraldehyde 3-phosphate dehydrogenase (GAPDH) were analyzed by RT-PCR as described in the Methods section. (**B**–**E**) MDA-MB-231 and RT-R-MDA-MB-231 cells were treated with TNF-α (10 ng/mL) (**B**,**D**) or ATP (100 μM) (**C**,**E**) for the indicated times. Then, NLRP3, NLRC4, ASC, cleaved caspase-1 (p10), and β-actin protein levels were analyzed by Western blotting (**B**,**C**). The band intensities were assessed by scanning densitometry (**D**,**E**). The values represent the mean ± standard error of the mean (SEM) of three independent experiments. * *p* < 0.05, ** *p* < 0.01, compared to the control MDA-MB-231 cells; ^#^
*p* < 0.05, ^##^
*p* < 0.01, compared to the control RT-R-MDA-MB-231 cells. CASP-1: caspase-1.

**Figure 2 ijms-21-03337-f002:**
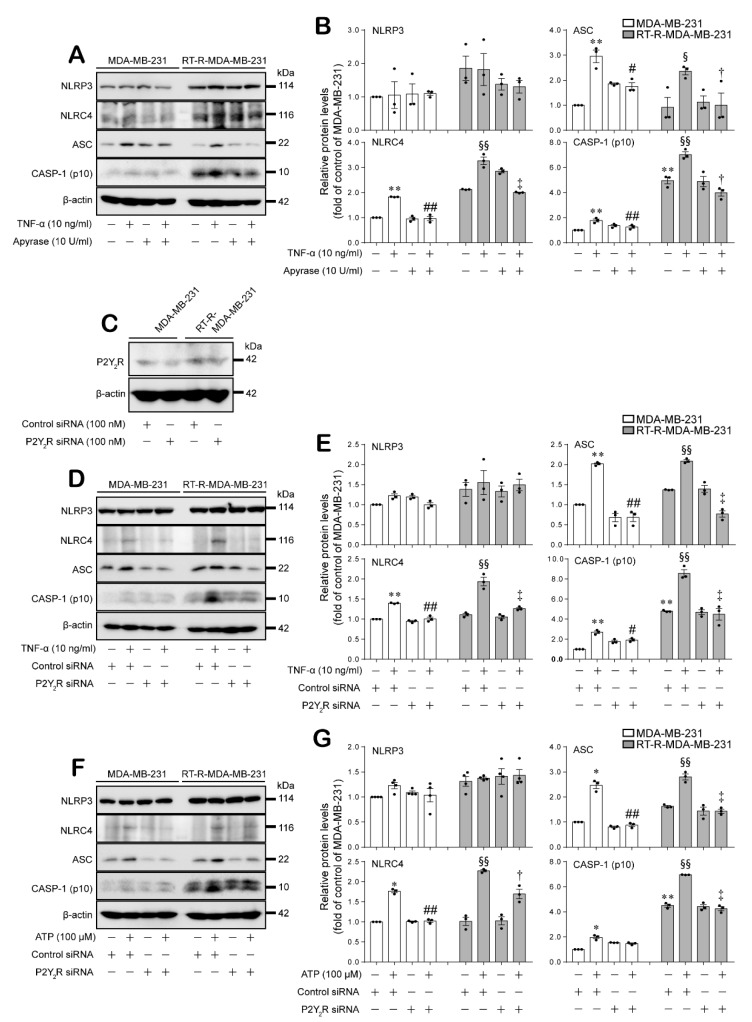
The inflammasome components NLRC4, ASC, and cleaved caspase-1 were induced by TNF-α or ATP in a P2Y purinergic receptor 2 (P2Y_2_R)-dependent manner in MDA-MB-231 and RT-R-MDA-MB-231 cells. (**A**,**B**) The cells were pretreated with apyrase (10 U/mL) for 1 h and then stimulated with TNF-α (10 ng/mL) for 12 or 24 h for the detection of cleaved caspase-1 (p10) or NLRP3, NLRC4, and ASC, respectively. The protein levels of inflammasome components were analyzed by Western blotting as described in the Methods section (**A**) and were quantified by densitometry (**B**). The values represent the mean ± SEM of three independent experiments. ** *p* < 0.01, compared to the control MDA-MB-231 cells; ^#^
*p* < 0.05, ^##^
*p* < 0.01, compared to TNF-α-treated MDA-MB-231 cells; ^§^
*p* < 0.05, ^§§^
*p* < 0.01, compared to the control RT-R-MDA-MB-231 cells; ^†^
*p* < 0.05, ^‡^
*p* < 0.01, compared to TNF-α-treated RT-R-MDA-MB-231 cells. (**C**) The cells were transfected with control or P2Y_2_R siRNA, and the knockdown efficiency was confirmed by Western blotting. CASP-1: caspase-1. (**D**,**E**) The control or P2Y_2_R siRNA-transfected cells were stimulated with TNF-α (10 ng/mL) for 12 or 24 h for the detection of cleaved caspase-1 (p10) or NLRP3, NLRC4, and ASC, respectively. The inflammasome component protein levels were analyzed by Western blotting as described in the Methods section (**D**) and were quantified by densitometry (**E**). The values represent the mean ± SEM of three independent experiments. ** *p* < 0.01, compared to the control MDA-MB-231 cells; ^#^
*p* < 0.05, ^##^
*p* < 0.01, compared to TNF-α-treated MDA-MB-231 cells; ^§§^
*p* < 0.01, compared to the control RT-R-MDA-MB-231 cells; ^‡^
*p* < 0.01, compared to TNF-α-treated RT-R-MDA-MB-231 cells. (**F**,**G**) Control or P2Y_2_R siRNA-transfected cells were stimulated with ATP (100 μM) for 3 or 24 h to detect NLRP3 and cleaved caspase-1 (p10) or NLRC4 and ASC, respectively. The inflammasome component protein levels were analyzed by Western blotting as described in the Methods section (**F**) and were quantified by densitometry (**G**). The values represent the mean ± SEM of three independent experiments. * *p* < 0.05, ** *p* < 0.01, compared to the control MDA-MB-231 cells; ^##^
*p* < 0.01, compared to ATP-treated MDA-MB-231 cells; ^§§^
*p* < 0.01, compared to the control RT-R-MDA-MB-231 cells; ^†^
*p* < 0.05, ^‡^
*p* < 0.01, compared to ATP-treated RT-R-MDA-MB-231 cells.

**Figure 3 ijms-21-03337-f003:**
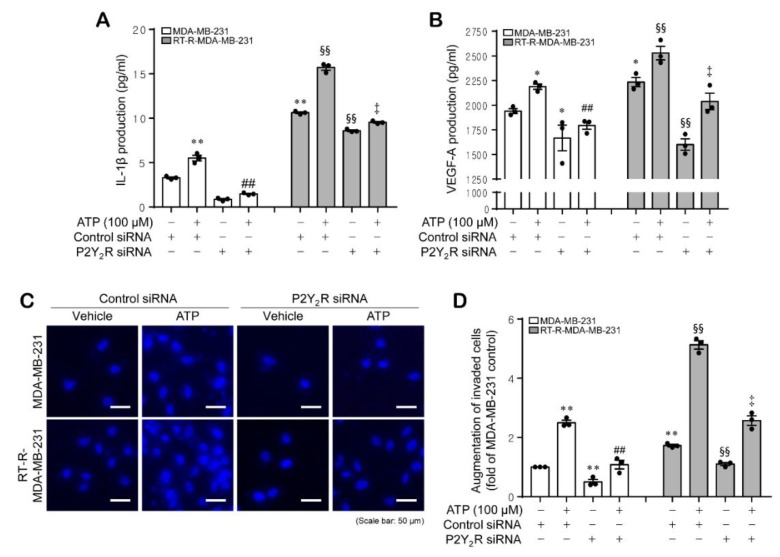
Inflammasome activation induced by ATP increased the secretion of interleukin (IL)-1β and vascular endothelial growth factor (VEGF)-A and the invasive ability of MDA-MB-231 and RT-R-MDA-MB-231 cells in a P2Y_2_R-dependent manner. (**A**,**B**) Control or P2Y_2_R siRNA-transfected cells were stimulated with ATP (100 μM) for 24 h. Then, IL-1β (**A**) and VEGF-A production (**B**) were measured as described in the Methods section. The values represent the mean ± SEM of three independent experiments. * *p* < 0.05, ** *p* < 0.01, compared to the control MDA-MB-231 cells; ^##^
*p* < 0.01, compared to ATP-treated MDA-MB-231 cells; ^§§^
*p* < 0.01, compared to the control RT-R-MDA-MB-231 cells; ^‡^
*p* < 0.01, compared to ATP-treated RT-R-MDA-MB-231 cells. (**C**,**D**) Control or P2Y_2_R siRNA-transfected cells were stimulated with ATP (100 μM) for 24 h. Then, Matrigel invasion assays were performed as described in the Methods section. The invaded cells that crossed the membrane were stained with 4′,6-diamidine-2′-phenylindole dihydrochloride (DAPI) (**C**) and counted in three randomly selected fields under a fluorescence microscope (**D**). The values represent the mean ± SEM of three independent experiments. ** *p* < 0.01, compared to the control MDA-MB-231 cells; ^##^
*p* < 0.01, compared to ATP-treated MDA-MB-231 cells; ^§§^
*p* < 0.01, compared to the control RT-R-MDA-MB-231 cells; ^‡^
*p* < 0.01, compared to ATP-treated RT-R-MDA-MB-231 cells. Scale bar: 50 μm.

**Figure 4 ijms-21-03337-f004:**
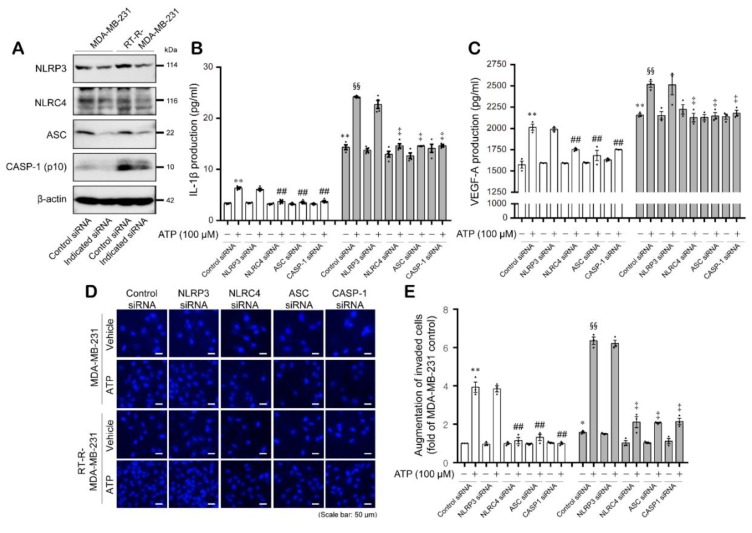
The inflammasome components NLRC4, ASC, and cleaved caspase-1, but not NLRP3, were involved in ATP treatment-enhanced secretion of IL-1β and VEGF-A and invasion of MDA-MB-231 and RT-R-MDA-MB-231 cells. (**A**) Cells were transfected with siRNA specific to each of the inflammasome components, and the knockdown efficiency was confirmed by Western blotting. CASP-1: caspase-1. (**B**,**C**) The indicated siRNA-transfected cells were stimulated with ATP (100 μM) for 24 h. Then, IL-1β (**B**) and VEGF-A production (**C**) were measured as described in the Methods section. The values represent the mean ± SEM of three independent experiments. * *p* < 0.05, ** *p* < 0.01, compared to the control MDA-MB-231 cells; ^##^
*p* < 0.01, compared to ATP-treated MDA-MB-231 cells; ^§§^
*p* < 0.01, compared to the control RT-R-MDA-MB-231 cells; ^‡^
*p* < 0.01, compared to ATP-treated RT-R-MDA-MB-231 cells. (**D**,**E**) The indicated siRNA-transfected cells were treated with ATP (100 μM) for 24 h. Then, Matrigel invasion assays were performed as described in the Methods section. The invaded cells that had crossed the membrane were stained with DAPI (**D**) and counted in three randomly selected fields under a fluorescence microscope (**E**). The values represent the mean ± SEM of three independent experiments. ** *p* < 0.01, compared to the control MDA-MB-231 cells; ^##^
*p* < 0.01, compared to ATP-treated MDA-MB-231 cells; ^§§^
*p* < 0.01, compared to the control RT-R-MDA-MB-231 cells; ^‡^
*p* < 0.01, compared to ATP-treated RT-R-MDA-MB-231 cells. Scale bar: 50 μm.

**Figure 5 ijms-21-03337-f005:**
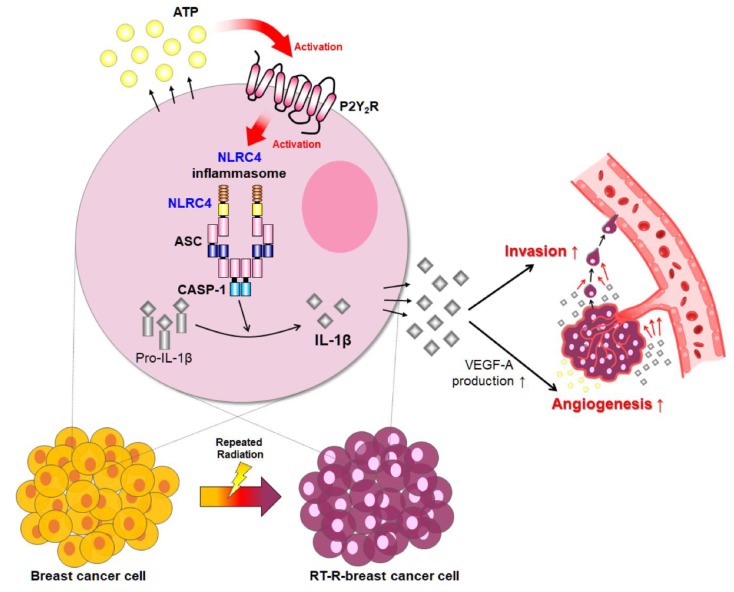
Schematic representation of the proposed role of P2Y_2_R in the regulation of the NLRC4 inflammasome and cancer cell progression in MDA-MB-231 and RT-R-MDA-MB-231 cells.

**Table 1 ijms-21-03337-t001:** RT-PCR primer sequences for inflammasome components.

Gene		Primer Sequence (5′-3′)	Annealing Temperature (°C)	Size (bp)
NLRP1	Forward	CCTGGTGGCTCAGTATGGGG	60	334
	Reverse	CTCATGGTCTGGGGAGCTTG		
NLRP3	Forward	GCACGTGTTTCGAATCCCAC	60	250
	Reverse	CCTGCTGGCTCCGGTGCTCC		
NLRP6	Forward	GGACGGACGCAGCATCCCGT	65	470
	Reverse	GGCCCTGCAGCACCACGGTC		
NLRC4	Forward	CAGAACCTGTCCTGTGGAGG	60	317
	Reverse	CCGCAGCTTCAGCAGCATGG		
AIM2	Forward	ACTCTTGCTAACAGGCCTGG	55	283
	Reverse	ACTTAGTGGCTTTGGTTTTG		
ASC	Forward	GCGCTGGAGAACCTGACCGC	65	210
	Reverse	CTCCTGCAGGCCCATGTCGC		
Caspase-1 (p10)	Forward	GAAAGCCCACATAGAGAAGG	55	218
	Reverse	CTCTTTCAGTGGTGGGCATC		
GAPDH	Forward	TCAACAGCGACACCCACTCC	60	126
	Reverse	TGAGGTCCACCACCCTGTTG		

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
