# Peer review of "NLRC4, ASC and Caspase-1 Are Inflammasome Components That Are Mediated by P2Y2R Activation in Breast Cancer Cells"

_ijms, 2020, doi:10.3390/ijms21093337_

Round 1

Reviewer 1 Report

The authors reported previously that  MDA RT-R breast cancer cells produce high levels of ATP, and ATP acts via P2Y purinergic receptor 2 (P2Y2R) to induce inflammasome activation to promote tumor invasiveness and tumor growth. In this manuscript, they have extended their initial findings to report on the identity of the inflammasome involved and the involvement of its components. They report that NLRC4 inflammasome and ASC and caspase 1 are induced by ATP acting via P2Y2R receptor. No significant differences between MDA and RTR-MDA cells were found except in the levels of IL-1 beta release to explain the higher invasive activity of RTR-MDA cells. The study is technically sound, and the results are clearly reported.

Comments:

  1. The rationale for studying MDA and RTR-MDA cells is not very clear and should be explained better in the introduction.
  2. The relevance of the observed differences between MDA and RTR-MDA cells in terms of their invasiveness and other properties should be discussed.
  3. Figs. 2 D and F NLRC4 and Casp 1 panels appear similar? Please verify.
  4. Do any of the treatments alter cell viability? Please indicate.
  5. How many cells were counted to analyze data shown in Figs. 3 C and D and Figs. 4 D and E? Please indicate in the figure legends.

Author Response

The authors reported previously that MDA RT-R breast cancer cells produce high levels of ATP, and ATP acts via P2Y purinergic receptor 2 (P2Y2R) to induce inflammasome activation to promote tumor invasiveness and tumor growth. In this manuscript, they have extended their initial findings to report on the identity of the inflammasome involved and the involvement of its components. They report that NLRC4 inflammasome and ASC and caspase 1 are induced by ATP acting via P2Y2R receptor. No significant differences between MDA and RTR-MDA cells were found except in the levels of IL-1 beta release to explain the higher invasive activity of RTR-MDA cells. The study is technically sound, and the results are clearly reported.

Comments:

1. The rationale for studying MDA and RTR-MDA cells is not very clear and should be explained better in the introduction.

→ Answer: Thank you for your comments. As you suggested, we reorganized the Introduction part and tried to explain the rationale and the study aim of this study (please see page 2, lines line 39~52 in red color). In addition, we added some sentences in the conclusion part as follows; “Moreover, as mentioned in the Introduction, inflammation has two aspects; inflammation is a defense mechanism to protect body from the damage, on the other hand, it also plays an important role in tumor promotion, malignant conversion and metastasis. Thus, to clarify the specific inflammasome components which are involved in the tumor progression and recurrence and to regulate those components could be an essential trial for cancer treatment. In this respect, our results suggest for the first time that P2Y2R is involved in activation of the NLRC4 inflammasome, which is composed of NLRC4, ACS and caspase-1, and highlight the importance of regulating P2Y2R activity to achieve a good prognosis in patients with RT-R tumors (Figure 5)” (page 11, lines 15~23).

2. The relevance of the observed differences between MDA and RTR-MDA cells in terms of their invasiveness and other properties should be discussed.

→ Answer: Thank you for your comments. To compare more clearly the differences between MDA and RTR-MDA cells observed in this study, we reorganized the discussion part (please see page 10~11 in red part) and inserted the following sentences (page 10, lines 20~25; “These results suggest that NLRC4, ASC and caspase-1 are inflammasome components that are mediated by P2Y2R activation in breast cancer cells. Even though this is common finding observed in both MDA-MB-231 and RT-R-MDA-MB-231, interestingly, these inflammasome components NLRC4, ASC and caspase-1 are more induced in RT-R-MDA-MB-231 than in MDA-MB-231 and are involved in the induced IL-1b secretion, VEGF-A production and invasion in RT-R-MDA-MB-231 cells”.

3. Figs. 2 D and F  D and F NLRC4 and Casp 1 panels appear similar? Please verify.

→ Answer: Thank you for your comments. Those blots look similar, but those are totally different blots obtained from different membranes. To make sure this point and help your understanding, we provide the original whole membranes. Please check them as shown below (please see attached file).

4. Do any of the treatments alter cell viability? Please indicate.

→ Answer: When we treated TNF-α or ATP, cell viabilities of the MDA-MB-231 and RT-R-MDA-MB-231 were not significantly changed at the observed time points and dose ranges used

5. How many cells were counted to analyze data shown in Figs. 3 C and D and Figs. 4 D and E? Please indicate in the figure legends.

→ Answer: As we mentioned in the Methods, we added MDA-MB-231 or RT-R-MDA-MB-231 cells (2 × 105 cells/500 ml) were added to each upper insert well. After 24 hr incubation, we counted the invaded cells which were stained with DAPI in three randomly selected field (same area) under a fluorescence microscope. As you indicated, we modified the Method section (4.7. Matrigel invasion assay; page 13 in red color) and added this point in the Figures 3 and 4 legends (in red color, page 8, Figure 3C-D legend; page 9, Figure 4D-E legend). Please check them.

Reviewer 2 Report

the authors investigated the role of NLRP3, NLRC4, ASC and caspase-1 mRNA in BC cells and radiotherapy-resistant BC cells.

I can't understand the main goal of the study. The authors demonstrated that in radiotherapy-resistant BC cells  there is an over-expression of  NLRP3, NLRC4, ASC and caspase-1 mRNA levels. The main concern is because the TNF-α or ATP treatment should be useful if NLRC4, ASC and caspase-1 are induced further. Namely, I expect that a correct therapy should reduce mRNA levels that are up-regulated in resistant-BCcells. this point eludes me. Could the authors explain better it?

Author Response

the authors investigated the role of NLRP3, NLRC4, ASC and caspase-1 mRNA in BC cells and radiotherapy-resistant BC cells.

I can't understand the main goal of the study. The authors demonstrated that in radiotherapy-resistant BC cells there is an over-expression of NLRP3, NLRC4, ASC and caspase-1 mRNA levels. The main concern is because the TNF-α or ATP treatment should be useful if NLRC4, ASC and caspase-1 are induced further. Namely, I expect that a correct therapy should reduce mRNA levels that are up-regulated in resistant-BC cells. this point eludes me. Could the authors explain better it?

→ Answer: Thank you for your comments. To make the aim of this study, we modified the Introduction part. Please see Page 2, lines 39~52 in red color.

Regarding TNF-α or ATP treatment, there seems to be a misunderstanding. As we mentioned in the Introduction, in our previous study, highly metastatic breast cancer cells MDA-MB-231 cells released high levels of ATP compared to lowly metastatic breast cancer cells, and the subsequent activation of P2Y purinergic receptor 2 (P2Y2R) promoted invasion and tumor growth by inducing inflammasome activation. Moreover, RT-R-MDA-MB-231 cells released higher levels of ATP than MDA-MB-231 cells and showed a more increased inflammasome activation in a P2Y2R-dependent manner. In addition, TNF-a which accumulate in the tumor microenvironment increased ATP released in MDA-MB-231 cells (Jin et al., 2014m Breast Cancer Res). Thus, in this study, we showed that NLRC4, ASC and cleaved caspase-1 were significantly increased by treatment with TNF-α or ATP in both MDA-MB-231 and RT-R-MDA-MB-231 cells (Figure 1B-E). TNF-α or ATP was not used as a therapy to reduce inflammasome components expression. Rather, TNF-α or ATP is a stimulant to induce inflammasome components expression through activation of P2Y2R.

Round 2

Reviewer 2 Report

the manuscript has been significantly improved